# Level of Vocabulary Development and Selected Elements Regarding Sensory Integration and Balance in 5-Year-Old Girls and Boys

**DOI:** 10.3390/children8030200

**Published:** 2021-03-07

**Authors:** Jacek Wilczyński, Grzegorz Ślęzak

**Affiliations:** 1Laboratory of Posturology, Collegium Medicum, Jan Kochanowski University in Kielce, Al. IX Wieków Kielc 19, 25–317 Kielce, Poland; 2Municipal Psychological and Pedagogical Clinic Complex, Kielce, 75–215 Koszalin, Poland; grzegorzslezak@wp.pl

**Keywords:** level of vocabulary development, selected elements of sensory integration and balance, Children’s Dictionary Test, Clinical Test of Sensory Integration and Balance

## Abstract

The aim of this research was to assess relationships between the level of vocabulary and selected elements of sensory integration and balance in 5-year-old girls and boys, showing the differences between them. The study group consisted of 290 5-year-old children (172 boys and 118 girls) with different levels of vocabulary development and selected disturbances in sensory integration and balance processes. To evaluate the developmental deficits of speech with regard to vocabulary, the Children’s Dictionary Test was used. The Clinical Test of Sensory Integration and Balance was also employed. In our research’s overall assessment, 118 children, i.e., 41%, had a low level of vocabulary, while 108 (37%) had an average level and 64 (22%) had a high level. However, the average score of all examined children (3.71 stens) indicates a low level of vocabulary development. Less developed vocabulary skills included the ability to create subordinate words and define concepts. There were no significant differences in the level of vocabulary between girls and boys. We observed disorders concerning selected elements of sensory integration and balance in most of the children, and more often in boys. There were statistically significant relationships between the level of vocabulary and selected disorders of sensory integration and balance; however, they were not unambiguous. Children with the lowest level of vocabulary in overall assessment obtained significantly the worst results in the Clinical Test of Sensory Integration and Balance (CTSIB) open eyes, hard surface test. However, in the closed eyes, hard surface test, the lowest score was obtained by children with a high overall assessment. In turn, in the open eyes, soft surface test, the lowest score was noted for children with average overall assessment. In the complex CTSIB test, the lowest score was achieved by children with low ability to define concepts. The problem of the relationship between vocabulary level of and sensory integration as well as balance requires further research. The demonstrated significant relationships between some aspects of vocabulary level and selected elements of sensory integration as well as balance confirm the need to care for the overall psychomotor sphere of a child.

## 1. Introduction

Language is the human ability to express and communicate desires, feelings, and thoughts through a system of specific signals, such as sounds, voice, written symbols, and gestures, whereas speech is an act of expression or the ability to describe desires, feelings, and thoughts in words or by voice communication. A child’s vocabulary, on the other hand, is the total words known to a child, depending on age, experience, intelligence, and language skills [1,2,3]. It is the basic component of systemic language competence, the essence of which is the ability to build an unlimited number of correct sentences from a limited number of words [4,5]. The development of a child’s vocabulary depends on many factors. Most often mentioned are environment, experience, parents’ education, access to culture, state of health, and mental fitness [6,7]. The development of each activity, including speech, is holistic; thus, there is a relationship between the development of a child’s speech and the processes of sensory integration and balance [8,9]. The processes of sensory integration and balance modulate stimuli reaching the child from the surrounding world. They also enable learning about the surrounding reality. 

In a child’s development, the ability to move and perform complex activities related to large and small motor skills is of particular importance [10,11]. Motor development affects the processes of thinking and processing stimuli reaching the child from the environment [12]. An important determinant is the condition of small motor skills, oral praxis, modulation of muscle tone throughout the body, and planning movements. Disorders in this regard can be observed before the first year of life. In most children, the above symptoms only mean transient developmental difficulties. However, some of them at age 6–7 may experience problems with modulating stimuli coming from the balance system. This can be seen by watching a child play and the inharmonious development of particular manipulative and motor skills [13].

The formation of sensory processes and the child’s balance begins during the prenatal period. Many external factors hinder the proper development of all senses and their integration. The uniqueness of each child means that each has a different, specific model of sensory processing, which modifies the reception of the world and significantly affects quality of life. Frequent environmental adaptation difficulties are caused by disturbances and delays in sensory development. Developmental deficits, also referred to as developmental dysfunctions, manifest themselves in disorders or delays of psychomotor development, and a slower rate of acquiring specific functions. Parallel deficits are distinguished, referring to a larger area of activity, e.g., delay in the development of large and small motor skills and deficits of fragmentary specificity, with a narrower scope, e.g., delay only regarding the motor skills of the hands with proper development of large motor skills [4,14,15].

Beginning their education, children often display irregularities in the reception and modulation of sensory stimuli. They concern muscle tension and postural reactions, having significant impact on group functioning and acquiring school skills [16]. Postural reactions regulate the stability of the body during rest or movement; they are a response to the task. The integration of proprioceptive, tactile, and visual information with directional information ensures the ability to maintain static and dynamic balance [17]. It also allows proper muscle tone, postural stability, stabilization of the field of vision, and coordination of movements. The modulation of impulses flowing from these sensory systems is subject to integration thanks to the vestibular nuclei of the brain [18,19]. Atrial nodes located in the brain stem have connections to the cerebellum, reticular formation, thalamus, and cerebral cortex. Disturbances in the integration of information transmission between brain structures may cause improper muscle tension (hypotonia or hypertonia), movement coordination disorders, abnormal muscle contraction related to resistance, abnormal balance responses, balance between flexion, and extension within different parts of the body [20,21]. Improper sensory integration may cause dysfunctions in vestibulospinal, vestibulo-oculic, peri-oculi-occlusion, and optic–ocular reactions, i.e., optokinetic reactions. Postural balance provides stability of the body despite movement, allowing development of precise movements of the head, limbs, eyeballs, and tongue. Disorders of this kind directly affect deficits in the sphere of speech transmission and reception. They also hinder learning and play (incorrect body posture, high level of fatigue, and reduced concentration) and negatively affect a child’s social and emotional functioning [22,23,24,25,26].

Although the relationship between processes of sensory integration and balance, as well as speech and vocabulary, seems obvious in scientific publications, the above problem is rarely analyzed. Previous studies have mostly been focused on the issues of speech and sensory integration, treating them separately [27,28,29,30]. In our research, however, we tried to establish a relationship between vocabulary development of children and the processes of sensory integration as well as balance. The aim of research was to assess relationships between the level of vocabulary and selected elements of sensory integration and balance in 5-year-old girls and boys, indicating differences between them. Hypothesis: We assumed that there is a significant relationship between the level of vocabulary development and selected elements of sensory integration and balance. The more disturbed the sensory integration processes, the lower the child’s vocabulary development.

## 2. Material and Methods

### 2.1. Patient Selection

The study group comprised 290 5-year-old children (172 boys and 118 girls). The examined children attended municipal kindergartens. The majority of children had a medium to high level of vocabulary development. Most of them experienced disorders regarding selected elements of sensory integration and balance. Inclusion criteria for research: 5-year-old children, girls and boys, the occurrence of selected disorders concerning sensory integration (SI) processes, consent to participate in examination. Criteria for exclusion from research: age below and over 5, physiological hearing disorders, intellectual disability, diseases and birth defects of the central nervous system. Before testing was carried out, the following documents were prepared: information for the subject containing detailed data on the objective and procedures of testing; parent/legal guardian consent form for the child’s participation in the study; statement of the parent/legal guardian for the processing of data related to participation in the study; and a statement from the researcher confirming informed consent for research from all parents/legal guardians of the children participating in the study. The research was carried out in the first half of 2016 at the Laboratory of Posturology, Collegium Medicum, Jan Kochanowski University. All research procedures were performed in accordance with the 1964 Declaration of Helsinki and with the consent of the University Bioethics Committee for Research at the Jan Kochanowski University in Kielce (Resolution No. 23/2015).

### 2.2. Outcome Measures

To assess the level of vocabulary development, the Children’s Dictionary Test was used. This test consists of 3 sub-tests: creating subordinate words, defining concepts, and creating superordinate words. The raw numerical results were converted into sten scores, and the development of vocabulary was expressed using the terms low, average, and high. Sub-test 1, creating subordinate words, comprised 10 questions: what toys do you know, what do children do, what colors do you know, what can you ride, what do your parents do, what are children like, what can you find in a house, what are people like, what can you play, and what do we eat and drink? The child had 10 min to answer, 1 min for each question. One point was given for each correct answer, which was a raw result. Sub-test 2, definition of terms, was similarly evaluated, and the raw result was converted into sten scores. For each correct answer, the child could receive 2 points. The test consisted of 10 questions: what is a ball, what are sleds, what is breakfast, what is milk, what is a hat, what is a kitchen, what is paint, what is a forehead, what is summer, and who is a leprechaun? For each question, 1 min was given for answering, approximately 10 min in total. Sub-test 3, the creation of superordinate words, was preceded by a preliminary exercise: “You probably know that some words can be replaced by different one, e.g., dog, horse, cat is...”. The time for answering each question was 30 s, and as before, there were 10 in total: doll, ball, building-block; bicycle, car, tractor; bread, sugar, soup; juice, tea, coffee; dress, shoes, trousers; room, kitchen, bathroom; red, white, black; write, read, counts; sweep, wipe, throw. For each correct answer, the child received 3 points. The raw numerical results were converted into sten scores, and vocabulary development was expressed using low (1–4 stens), average (5–6 stens), and high (7–10 stens) scores. The higher the value in the test, the higher the level of vocabulary development. Research was conducted at a speech therapy office. The entire examination lasted, depending on the child’s psychophysical abilities, about 30 min [31].

The Clinical Test of Sensory Integration and Balance was employed. This is a standardized clinical test performed on the Biodex Balance System static platform. The Clinical Test of Sensory Integration and Balance (CTSIB) is effective in identifying people with varying degrees of sensory integration and balance disturbances. CTSIB consists of 4 stages and provides general insight into the integration of various senses to improve balance and compensation when the activity of one or several senses is impaired. The following tests were performed: with open eyes on hard surface, a combination of visual, vestibular, and somatosensory elements; with closed eyes on a hard surface, removal of the visual element to assess the function of vestibular and somatosensory elements; with open eyes on a soft surface, used to assess somatosensory and visual interaction; and with closed eyes on a soft surface, used to assess somatosensory interaction and balance. The lower the value in the test, the fewer the disturbances regarding selected elements of sensory integration and balance [32].

In order to precisely select the research group, rigorous exclusion criteria for individuals with a physiological hearing disorder were adopted. In these tests, a screen auditory tonometer (SAT) with central tests was used. The device enabled precise assessment of a child’s hearing organ efficiency. It consisted of modules for peripheral hearing and central auditory processing diagnosis. Performing the test did not require a silenced chamber. Participants in the study underwent diagnosis regarding peripheral hearing efficiency at 125 Hz, 250 Hz, 500 Hz, 750 Hz, 1.5 kHz, 2 kHz, 3 kHz, 4 kHz, 6 kHz, and 8 kHz [33].

### 2.3. Statistical Analysis

Statistical analyses were performed with the IBM SPSS PL v.22 statistical package (Predictive Solutions, Cracow, Poland). Normality of distribution of interval variables was assessed using the Shapiro–Wilk test. In the studied group, the analyzed variables did not meet the criteria for normality of distribution in the subgroups in most cases. In addition, the compared groups of children differed in numbers. For this reason, non-parametric tests based on rank were used to assess the significance of differences: the Mann–Whitney *U* test to evaluate the significance of differences between 2 independent groups or the Kruskal–Wallis test to assess the significance of differences between 3 or more compared independent groups. Statistically significant differences were at the level of *p* < 0.05.

## 3. Results

### 3.1. Characteristics of the Level of Vocabulary Development

The average result of the whole test (overall assessment) was 3.71 stens—a low level of vocabulary development. The highest individual score was 6 stens, and the lowest was 1 sten. In the creating subordinate words sub-test, the average sten score was 3.28—a low level. The diversity of individual results ranged from a low sten score of 1 to a high level of 7 stens. In the second sub-test, defining concepts, the average sten score was 4.76—a low level—while the highest individual score was 10 stens, and the lowest was 1 sten. In the creating superordinate words sub-test, the subjects achieved an average score of 7.18 stens, i.e., a high level. The differentiation of the subjects was significant from the highest individual result at the level of 10 stens to the lowest of 1 sten (Table 1). Analyzing the data in groups according to gender, a similar trend was observed: The boys achieved a low score of the whole test (overall assessment) of 3.74 stens, 0.03 stens higher than the overall score for all subjects; the girls presented a low level of 3.66 stens, lower than the overall score by 0.05 and the score for boys by 0.08 stens. In the sub-test regarding the creation of subordinate words, the boys achieved a low score of 3.28 stens, identical to the total; a similar result was achieved by girls, totaling 3.29 stens, only 0.01 stens better than the boys. In the defining concepts sub-test, the boys achieved an average score of 4.83 stens—low, but slightly better than the general population by 0.07 stens and slightly better than the girls by 0.16 stens, whose average was 4.67 stens. In the creating superordinate words sub-test, the boys also achieved a high level of 7.16 stens, a score lower than the general average result by only 0.02 stens, and lower than the average score for the girls (7.19) by 0.03 stens. In the overall assessment, 118 children, i.e., 41%, had a low vocabulary level, while 108 (37%) had an average level and 64 (22%) had a high level. In the creating subordinate words test, 120 children (41%) had a low vocabulary level, while 106 (37%) had an average level and 64 (22%) had a high level. In the defining concepts test, 121 children, equaling 42%, had a low level of vocabulary, while 99 (34%) had an average level and 70 (24%) had a high level. In the creating superordinate words test, 123 children, i.e., 42%, had a low level of vocabulary, while 97 (34%) had an average level and 70 (24%) had a high level. Nonetheless, no statistically significant differences were noted in the level of vocabulary between girls and boys (Table 1).

### 3.2. Results of the Clinical Test of Sensory Integration and Balance (CTSIB)

In the open eyes, hard surface test, the boys’ average score was 2.89, and for girls it was 2.61. No significant differences were observed here (*p* = 0.081). In the closed eyes, hard surface test, the boys’ average score was 2.66, while the girls’ score was 2.30. The girls obtained a better result here (*p* = 0.018). In the open eyes, soft surface test, the boys ‘average score was 3.11, and the girls’ was 2.78. Here, the girls also obtained a better result (*p* = 0.007). In the closed eyes, soft surface test, the average score for boys was 3.82, and among girls it was 3.54. Here too, the girls obtained a better result (*p* = 0.013). In overall assessment, the average score for boys was 3.11, and for girls it was 2.80. In this case, the girls also obtained a better result (*p* = 0.002) (Table 2).

### 3.3. The Level of Vocabulary Development and Selected Elements of Sensory Integration and Balance

There were statistically significant relationships between vocabulary level and selected disorders of sensory integration and balance, but they were not unequivocal. Children with the lowest level of vocabulary in the overall evaluation obtained significantly worse results in the CTSIB open eyes, hard surface test (*p* = 0.01). On the other hand, in the closed eyes, hard surface test, the lowest score was obtained by children with a high overall rating of vocabulary (*p* = 0.04). In contrast, in the open eyes, soft surface test, the lowest score was obtained by children with an average overall vocabulary rating (*p* = 0.03). In the complex CTSIB test, the lowest result was obtained by children with low defining concepts (*p* = 0.04) (Table 3).

## 4. Discussion

In our research, we tried to determine the relationship between vocabulary level and selected elements of sensory integration and balance. It should be noted that the studied children were to soon start school education, and a low level of vocabulary development is a significant obstacle in the functioning of both the role of pupils and peers. Children with limited vocabulary development are not able to meet the social, physical, motivational, emotional, cognitive, or verbal requirements of school. Most often, they build short sentences with poor syntactic structure. Children whose vocabulary is poor not only achieve worse academic results but also fall into different conflicts with their peers or isolate themselves. Their negative social behavior may result from the unmet need to provide information. Speech, in which the vocabulary resource is an important component, becomes the main means of interaction understood as the exchange of values and information on which physical and emotional existence depends. A child, honestly and authentically, communicates what he or she feels to others. The ability to directly express one’s own experiences provides a sense of authenticity, ensuring autonomy and identity, and puts a child in situations where he or she feels like an author of his/her own behavior. Within this context, the issue of these studies is fully justified [34].

Children with vocabulary development disorders have been in the field of research and therapeutic interest for many years. Research by Morgan et al. [27], using quantitative imaging of the brain to observe the activity of neural networks, indicates that the basis of speech development disorders and sensory processes is common. These authors, using magnetic resonance imaging, examined 20 preschool children. The exclusion of children with neurological deficits and intellectual disability from the study group could be used to formulate conclusions on the importance of MRI variability and diversity of phenotypes in the study participants. Children with speech disorders had structural and functional anomalies in the left supramarginal gyrus and functional anomalies in the posterior bilateral cerebellum. Increased anomaly and radiant diffusivity of the left arcuate fasciculus was also shown, and the cortical and subcortical surfaces of the neuronal network were disturbed. In these studies, it was also indicated that there is a need to further explore the problem in order to solve the neurobiological background of speech disorders. It is important to define individual speech markers and language abilities [27]. 

In our research, it is shown that in the overall assessment, 118 children, i.e., 41%, had a low level of vocabulary, while 108 (37%) had an average level and 64 (22%) had a high level. However, the average score of all examined children (3.71 stens) indicates a low level of vocabulary development. In the creating subordinate words test, 120 children (41%) had a low vocabulary level, while 106 (37%) had an average level and 64 (22%) had a high level. In the defining concepts test, 121 children, that is 42%, had a low level of vocabulary, while 99 (34%) had an average level and 70 (24%) had a high level. In the creating superordinate words test, 123 children, totaling 42%, had a low level of vocabulary, while 97 (34%) had an average level and 70 (24%) had a high level. However, there were no statistically significant differences in the level of vocabulary between girls and boys. In the CTSIB open eyes, hard surface test, the boys’ average score was 2.89, and for girls it was 2.61. No significant differences were observed in this case (*p* = 0.081). In the closed eyes, hard surface test, the boys’ average score was 2.66, while the girls’ was 2.30. The girls obtained a better result here (*p* = 0.018). In the open eyes, soft surface test, the boys’ average score was 3.11, while the girls’ was 2.78. In this instance too, girls obtained a better result (*p* = 0.007). In the closed eyes, soft surface test, the average score obtained by boys was 3.82, and for girls it was 3.54. Here, the girls also achieved a better result (*p* = 0.013). In our research, there were statistically significant relationships between vocabulary level and selected disorders of sensory integration as well as balance, but they were not unequivocal. Children with the lowest level of vocabulary in the overall evaluation obtained significantly worse results in the CTSIB test, open eyes, hard surface (*p* = 0.01). On the other hand, in the closed eyes, hard surface test, the lowest score was obtained by children with a high overall rating of vocabulary (*p* = 0.04). In contrast, in the open eyes, soft surface test, the lowest score was obtained by children with an average overall vocabulary rating (*p* = 0.03). In the complex CTSIB test, the lowest result was obtained by children characterized by low defining concepts (*p* = 0.04).

In other studies [28], including 524 girls and 528 boys at the age of 3, only 157 children (14.92%) had low vocabulary skills; 358 (34.03%) were at an average level of vocabulary efficiency, while a high level was attained by 537 children (51.04%). It was also confirmed in subsequent studies that there is a relationship between sensory processes and balance as well as the level of a 3-year-old child’s vocabulary [29]. The most disturbed areas of sensory integration were the dynamic balance of 109 subjects (71%), and the static balance of 103 subjects (67%), while a low level of vocabulary was represented by 67 subjects (43.79%). A low vocabulary level was obtained by 129 subjects (84.31%); the highest level of ability to create subordinate words was achieved by 136 subjects (88.9%) and defining concepts by 93 subjects (60.78%). In the creation of superordinate words, a low level was noted in 72 subjects (42%) [29]. A positive result indicating no SI disorder among boys was recorded in three trials: eye–hand preference—68%, Symmetrical tonic neck reflex—71.5%, and gravitational danger—70.9%. In the case of girls, results indicating the absence of disorders was recorded in four trials: eye–hand preference—68.8%, STOS—62.7%, gravitational danger—63.6%, and trunk stabilization—57.6%. Therefore, it can be stated that these SI processes are not crucial in the development of vocabulary. On the other hand, children from the study group had such disturbances in sensory integration processes that were not found in the control group. Consequently, it may be assumed that it is these areas of sensory integration that are most closely related to the acquisition of vocabulary. This concerned motor planning, static and dynamic balance, and postural reactions—100% of boys. In girls, this involved postural mechanisms—63.6%—and motor planning—30% [29]. A similar condition was also confirmed by speech tests among 112 children (36 girls and 76 boys) aged 6-8 years performed by Jodzis [35]. In this research, the aspect of vocabulary appeared, analyzed as one of the sub-tests, and was low in 49 children (54%). There were also differences in the vocabulary of 6-year-old boys and girls. The general vocabulary development of girls turned out to be higher than boys. An analogous tendency was also recorded in vocabulary skills, i.e., in creating subordinate words, defining concepts, and creating superordinate words [35]. Many authors pointed to the variety in mastery of vocabulary by girls and boys [36,37,38,39]. Much research has been conducted on gender differences in language development. As early as 1922, Jespersen proved that younger girls, as a rule, learn to speak earlier and faster than boys and outperform them in correct pronunciation. The 1954 research by McCarthy showed that there is “a subtle difference in favor of girls regarding all aspects of language studied”. Garai and Scheinfeld also wrote in 1968 that girls outperformed boys in verbal fluency, correct use of language, sentence complexity, grammatical structure, spelling, and articulation [40]. In a different study, the correlation between balance ability and speech–language development in children was observed [41]. The longitudinal study was conducted in order to establish whether the success rate of reflexes related to maintaining balance at birth is in correlation with the success rate of maintaining balance in early childhood, as well as to examine the correlation of a certain level of speech and language development with the ability of maintaining balance at birth and at the age of 5. The main study group included 54 children of both genders, aged 5.0 to 5.4, whose balance ability and speech and language status were evaluated based on a battery of standardized tests, whereas the group of reflexes related to the function of the vestibular sense was clinically tested on the third day after birth, within the same sample of children. The data at birth and at the age of 5 were recorded by means of a digital camera then scored and statistically and descriptively processed. The research results indicated a statistically significant correlation between the achieved level of balance ability in the newborns and 5-year-olds, as well as between balance skills and a certain level of speech and language development in children at the age of 5. The importance of this research lies in new knowledge in the domain of maturation of vestibular function immediately after birth, given that this segment of physiology of a newborn has not been processed in such a way so far, as well as in the recognition of function of the vestibular sense as another parameter of a child’s maturation [41].

Clinical sensory integration tests are rarely found in scientific publications. The interesting research by Lekskulchai et al. [42] involved the CTSIB test being used to determine postural deviations and the strategy of the motion analysis system as the gold standard. Six clinical trials of the CTSIB were carried out. The study group consisted of 17 children with an average age of 9.34. The sensitivity of the standard measure to detect immature movement strategies ranged from 62.96% to 75.71%, while the specificity of the data obtained ranged from 68.12% to 97.22%. Positive and negative predictive values ranged from 46.43% to 94.74%. This study was capable of detecting immature motor strategy on stable and unstable surfaces [42]. It may be concluded that the cited studies carried out by other authors generally confirm our observations regarding the relationship between sensory integration and balance processes as well as speech development in terms of vocabulary. There is no contradiction here. In the overall assessment, 118 children, i.e., 41%, had a low vocabulary level, 108 children (37%) had a medium level, while 64 children (22%) had a high level. The percentage of children with medium (37%) and high (22%) levels of vocabulary was 59% (the majority). However, in sten scores, the average for the entire study group was 3.7, a low vocabulary level but on the verge of the average level. This results from the ranges; the raw numerical results were converted into sten scores, and the development of vocabulary was expressed using low (1–4 stens), average (5–6 stens), and high (7–10 stens) scores. The overall mean level of vocabulary in sten scores was lowered by the lower values of the ranges obtained in the group with the medium and high vocabulary levels. The limitation of our research was, inter alia, no assessment of the integration concerning other senses (taste, smell, touch, etc.). In the future, in addition to examining the vestibulospinal reactions, it will also be necessary to analyze the vestibulo-ophthalmic reactions, periocular occlusion and visual–eye reactions, i.e., optokinetic reactions, in connection with the development of a child’s vocabulary.

## 5. Conclusions

The mean score of all the examined children (stens) indicates a low level of vocabulary development. Less developed vocabulary skills were related to creating subordinate words and defining concepts. There were no significant differences in the level of vocabulary between girls and boys. We observed disorders concerning selected elements of sensory integration and balance in most children, more often in boys. There were statistically significant relationships between the level of vocabulary and selected disorders of sensory integration and balance, but they were not unambiguous. Children with the lowest level of vocabulary in the overall assessment obtained significantly worse results in the CTSIB open eyes, hard surface test. However, in the closed eyes, hard surface test, the lowest score was obtained by children with high overall assessment. In turn, in the open eyes, soft surface test, the lowest score was obtained by children with average overall assessment. In the complex CTSIB test, the lowest score was obtained by children with low ability to define concepts. The problem of the relationship between the level of vocabulary and sensory integration, as well as balance, requires further research. The demonstrated significant relationships between some aspects of the vocabulary level and selected elements of sensory integration and balance confirm the need to care for the overall psychomotor sphere of a child.

## Figures and Tables

**Table 1 children-08-00200-t001:** Results of the Dictionary Test for the studied children.

Children’s Dictionary Test standardisation						
Sub-Tests	M		SD	Med	25 Percen	75 Percen
Creating subordinate words	3.28		2.40	3.00	1.00	5.00
Defining concepts	4.76		2.78	5.00	2.00	7.00
Creating superordinate words	7.18		2.27	7.00	6.00	9.00
Overall assessment	3.71		2.25	4.00	1.00	5.00
Dictionary Test results for girls and boys						
Sub-Tests	Gender	M	SD	Med	25 Percen	75 Percen
Creating subordinate words	Boys	3.28	2.08	3.00	1.00	5.00
	Girls	3.29	2.22	3.00	1.00	5.00
Defining concepts	Boys	4.83	2.69	5.00	3.00	7.00
	Girls	4.67	2.92	4.00	2.00	7.00
Creating superordinate words	Boys	7.16	2.27	7.00	6.00	9.00
	Girls	7.19	2.29	7.00	6.00	9.00
Overall assessment	Boys	3.74	2.23	4.00	1.00	5.00
	Girls	3.66	2.30	3.00	1.00	5.00

**Table 2 children-08-00200-t002:** Results of the trials for the Clinical Test of Sensory Integration and Balance.

Variables	Gender	M	SD	Median	*p*
Open eyes, hard surface	Boys	2.89	1.56	2.52	0.081
	Girls	2.61	1.38	2.12	
Closed eyes, hard surface	Boys	2.66	1.30	2.37	0.018
	Girls	2.30	1.03	2.09	
Open eyes, soft surface	Boys	3.11	1.20	2.89	0.007
	Girls	2.78	0.99	2.77	
Closed eyes, soft surface	Boys	3.82	1.18	3.86	0.013
	Girls	3.54	1.00	3.47	
Overall assessment	Boys	3.11	0.98	3.02	0.002
	Girls	2.80	0.77	2.65	

**Table 3 children-08-00200-t003:** Clinical Test of Sensory Integration and Balance in children with varied vocabulary development.

**Variables**	**Verbal Assessment**	**Open Eyes, Hard Surface**					**Closed Eyes, Hard Surface**				
		**N**	**M**		**SD**	***p***	**M**		**SD**		***p***
Creating subordinate words	Low	120	2.93		1.58	0.28	2.47		1.22		0.19
	Average	106	2.61		1.37		2.46		1.23		
	High	64	2.78		1.52		2.69		1.17		
Defining concepts	Low	121	2.86		1.58	0.80	2.66		1.21		0.08
	Average	99	2.68		1.44		2.37		1.10		
	High	70	2.76		1.43		2.47		1.35		
Creating superordinate words	Low	123	2.79		1.63	0.35	2.46		1.22		0.51
	Average	97	2.61		1.26		2.46		1.01		
	High	70	2.99		1.54		2.70		1.43		
Overall assessment	Low	118	3.02		1.54	0.01	2.51		1.28		0.04
	Average	108	2.46		1.41		2.34		1.01		
	High	64	2.85		1.48		2.83		1.35		
**Variables**	**Verbal Assessment**	**N**	**Open Eyes, Soft Surface**			**Closed Eyes, Soft Surface**			**Complex Result**		
			**M**	**SD**	***p***	**M**	**SD**	***p***	**M**	**SD**	***p***
Creating subordinate words	Low	120	2.88	1.01	0.70	3.53	1.01	0.06	2.94	0.92	0.76
	Average	106	3.08	1.20		3.93	1.23		3.00	0.91	
	High	64	3.00	1.21		3.53	1.01		3.03	0.91	
Defining concepts	Low	121	3.03	1.06	0.52	3.88	1.12	0.09	3.09	0.83	0.04
	Average	99	2.96	1.18		3.58	1.02		2.89	0.90	
	High	70	2.92	1.19		3.57	1.22		2.93	1.05	
Creating superordinate words	Low	123	2.94	1.12	0.72	3.70	1.14	0.34	2.93	1.05	0.94
	Average	97	2.97	1.08		3.78	1.01		2.96	0.90	
	High	70	3.06	1.22		3.61	1.23		2.96	0.74	
Overall assessment	Low	118	2.88	1.01	0.57	3.53	0.97	0.03	2.98	0.91	0.67
	Average	108	3.00	1.18		3.91	1.28		2.91	0.83	
	High	64	3.12	1.23		3.67	1.06		3.12	1.02	

## Data Availability

The data presented in this study are available on request from the corresponding author.

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
