# Peer review of "Level of Vocabulary Development and Selected Elements Regarding Sensory Integration and Balance in 5-Year-Old Girls and Boys"

_children, 2021, doi:10.3390/children8030200_

Round 1
Reviewer 1 Report
General comment:
First of all, I would like to say that I am very thankful to have the opportunity to read this study. The suggestions given in this document are intended to improve your work. If you do not agree with any of them, please explain them to me, and we will try to reach a consensus. The same feedback document will be given to both editors and authors.
I think this article is relevant, innovative and useful for both educational and health professionals, but there are issues that I think should be addressed:
Introduction section:
- On line 37 the bracket must be closed.
- The statement in lines 62-64 should be supported by references.
- In relation to the dictionary, it might be interesting to add expected aspects of its development in children around the age group analysed.
- Sensory integration is about how a person is able to integrate the information that is received from the eight senses: sight, hearing, smell, taste, touch, proprioception, vestibular and interoception. Dysfunctions in sensory integration can cause difficulties far more diverse than those described. https://ajot.aota.org/article.aspx?articleid=1866929. They only address vision, proprioception and vestibular, those most intimately related to balance. If the authors are just looking at the relationship of certain senses with balance, I think this aspect should be clearer, otherwise it could be misleading (this comment applies to all sections of the paper).
Methods section:
- Lines 95-97 “The study group consisted of 290 5-year-old children (172 boys and 118 girls) with speech development deficits in terms of dictionary and disturbances of sensory integration and balance processes” How was this assessed?
- Please explain more clearly the inclusion and exclusion criteria.
- Line 132-133 “The method of assessing sensory integration and balance was the Clinical Test of 132 Sensory Integration and Balance”. Please add the reference to the validation study or the manual for this test.
- Line 134 Remember to explain all acronyms.
- Statistical analysis: Please explain in more detail and include references to the statistical software, statistical test and criteria used.
- I think it would be advisable to report the statistical power and provide a statistic for the magnitude of effect
Results section:
- I think it would be a good idea to simplify the tables, providing the mean and standard deviation are enough.
- Please explain all abbreviations in the table footer.
Author Response
I am submitting a revised version of the article titled "The level of Dictionary Development and Selected Elements Regarding Sensory Integration and Balance in 5-year-old Girls and Boys” to be considered for publication in Children. I confirm that this work is original and has not been previously published, nor is it currently under consideration for publication elsewhere. The problem addressed in this work is relevant to the scope of your journal. All procedures performed in studies involving human participants were in accordance with the ethical standards of the institutional and/or national research committee and with the 1964 Declaration of Helsinki and its later amendments or comparable ethical standards. We would like to thank the Reviewers for the time and effort put into the review of our manuscript. It was quite a challenge for us to answer some of the greatly insightful and very detailed questions. However, we have made effort to meet this challenge. We hope that after undergoing revision and following extensive changes, the article is interesting and will be accepted for publication. Below, please find a detailed, point-by-point description of the changes applied in the text as well as responses to comments.
Response to comments by Reviewer 1. Comments
First of all, I would like to say that I am very thankful to have the opportunity to read this study. The suggestions given in this document are intended to improve your work. If you do not agree with any of them, please explain them to me, and we will try to reach a consensus. The same feedback document will be given to both editors and authors. I think this article is relevant, innovative and useful for both educational and health professionals, but there are issues that I think should be addressed:
Point 1:Introduction section:On line 37 the bracket must be closed. The statement in lines 62-64 should be supported by references.In relation to the dictionary, it might be interesting to add expected aspects of its development in children around the age group analysed. Sensory integration is about how a person is able to integrate the information that is received from the eight senses: sight, hearing, smell, taste, touch, proprioception, vestibular and interoception. Dysfunctions in sensory integration can cause difficulties far more diverse than those described. https://ajot.aota.org/ article.aspx?articleid=1866929. They only address vision, proprioception and vestibular, those most intimately related to balance. If the authors are just looking at the relationship of certain senses with balance, I think this aspect should be clearer, otherwise it could be misleading (this comment applies to all sections of the paper).
Response 1:
In line 37, the parenthesis (to footnote [6,7]) has been closed. The statement in lines 62-64 has been supported by literature. So far, studies have mostly been focused on the issues of speech, sensory integration and balance, treating them separately. Sensory integration was most often studied by classical clinical observation. According to Ayres (17 attempts to evaluate sensory integration processes). This tool is sometimes criticised for its lack of objectivity and reliability, which is why in our research, we used the modern Biodex Balance System platform on which a test called Clinical Test of Sensory Integration and Balance was performed. This platform does not allow for the assessment of all 8 senses (sight, hearing, smell, taste, touch, proprioception, vestibule and interoception). This is one of the reasons why we changed the title of the article and its purpose. Title: The level of Dictionary Development and Selected Elements Regarding Sensory Integration and Balance in 5-year-old Girls and Boys. Purpose: The aim of the research was to assess the relationship between the level of the dictionary and selected elements of sensory integration and balance in five-year-old girls and boys.
Point 2: Methods section:Lines 95-97 “The study group consisted of 290 5-year-old children (172 boys and 118 girls) with speech development deficits in terms of dictionary and disturbances of sensory integration and balance processes” How was this assessed?Please explain more clearly the inclusion and exclusion criteria.Line 132-133 “The method of assessing sensory integration and balance was the Clinical Test of 132 Sensory Integration and Balance”. Please add the reference to the validation study or the manual for this test. Line 134 Remember to explain all acronyms.Statistical analysis: Please explain in more detail and include references to the statistical software, statistical test and criteria used.I think it would be advisable to report the statistical power and provide a statistic for the magnitude of effect.
Response 2: Response 2: Line 95-97: The study group consisted of 290, 5-year-old children (172 boys and 118 girls) with different levels of dictionary development and selected disturbances of sensory integration and balance processes. Selected sensory integration disorders, assessed with the modern Biodex Balance System platform, on which so-the called Clinical Test of Sensory Integration and Balance was performed. Inclusion criteria for research: 5 years of age, girls and boys, the occurrence of selected disorders of sensory integration (SI) processes, consent for the examination. Criteria for exclusion from research: age below or over 5, physiological hearing disorders, intellectual disability, diseases or birth defects of the central nervous system. Line 132-133: I have provided references [Moran RN, Cochrane G. Preliminary study on an added vestibular-ocular reflex visual conflict task for postural control. Concussion. 2020; 14; 5 (2): CNC73. doi: 10.2217/cnc-2020-0003. Line 134: Statistical analyses were performed with the IBM SPSS PL v.22 statistical package. Normality of distribution concernign interval variables was assessed using the Shapiro-Wilk test. In the studied group, the analysed variables did not meet the criteria for normality of distribution in the subgroups in the majority of cases. In addition, the compared groups of children differed in numbers. For this reason, non-parametric tests based on rank were used to assess the significance of differences: Mann-Whitney U test to evaluate the significance of differences between 2 independent groups or the Kruskal-Wallis test to assess the significance of differences between 3 or more compared independent groups. Statistically significant differences were at the level of p<0.05.
Point 3: I think it would be a good idea to simplify the tables, providing the mean and standard deviation are enough. Please explain all abbreviations in the table footer.
Response 3: The table has been simplified and all abbreviations are explained in all the tables.
Once more, we are exceptionally grateful for your in-depth view of our article. Your insight and comments will definitely allow for an increase in the substantive value of the manuscript. We hope that our detailed responses and the extensive changes to the text are sufficient for the publication of our text. Thank you for your devoting you time and effort.
Yours sincerely,
Assoc. Prof. UJK Jacek Wilczyński, Ph.D.

Reviewer 2 Report
Dear authors, Dear editors,
Thanks very much for asking me to review the manuscript “Speech development deficits in terms of dictionary and processes of sensory integration and balance in five-year-old children”, submitted for publication in Children. The authors have conducted a study to evaluate the relationship between speech deficits in terms of dictionary as well as sensory integration and balance in children.
The study is overall interesting, however, after careful review, I am advising the editors to re-consider the manuscript for publication after major revisions and resubmission by the authors. Introduction, methods and discussion should be extensively refined and improved.
Many regards
INTRODUCTION
-L26. The authors make a differentiation between language and speech. If they consider that they have to make such a differentiation, they must take into account that language is the ability of humans to express and communicate desires, feelings and thoughts through a system of particular signal like sounds, voice, written symbols and gestures. While the speech I the act of the expressing or the faculty of describing desires, feelings and thoughts by words or vocal communication.
-L41. The sentence “Motor development affects the processes of thinking and processing stimuli reaching the child from the environment” would benefit from a supporting reference.
-L46. The authors should concrete the sentence “At the age of 6, a child affected by developmental deficits will have problems…” There are no defined the “developmental deficits”.
In the discussion appear a series of research that address the aim of this investigation but that do not appear in the introduction. These investigations should be mentioned in the introduction. Also, all investigations that appear in the introduction are not discussed in the discussion section. It does not appear that there is a relationship between the state of the art and the discussion of the results of this research.
The authors must define what they understand by children with speech deficits: SLI, phonological disorders, developmental co-ordination disorder (dispraxya), etc.
The aim is to assess the relationship between speech deficits regarding dictionary and sensory integration balance in 5-year-old children without differences by sex. However, the results are explained in terms of boys and girls. The aim should be to rewrite.
On the other hand, the authors do not explain the hypothesis or hypotheses.
METHODS
There are several issues in the method that need to be carefully addressed.
-The patient selection: The description of the participants is scarce. We do not know if all the participants have development deficits of speech and disturbances of sensory integration and balance processes at the same time, what kind of speech development deficits have the participants. The manuscript does not explain if the participants attend a special education school, who has diagnosed them, what is the degree of speech development deficits and disturbances of sensory integration and balance processes. They authors could provide more information on participants and define them specifically because we do not know if they have speech delay, language delay or SLI.
- Outcome measures:
L113: It is important that authors explain what the range of scores “low, average and high” to interpret the results.
In addition, the authors must report on how the CTSIB is scored.
In the procedure, it would be interesting for the authors to explain how the evaluation was, individually, at the participants' home, in the hospital office, how long did the assess last with each participant, firstly, which they assessed first, the language component or the motor component, etc.
RESULTS
-L209. Please, consider rearranging the results. First, they could explain the total results and then the results by gender. Second, the total results in the subtest and then the results by gender in them.
In this way they avoid the repeated results that appear in the different lines (i.e., L167 and L177, 3.71)
The authors should change the abbreviatures of mean (X) and standard deviation (s). Mean= M and Standard Deviation= SD. The p-values must be written in lowercase in the Table 2.
L185: The Table is the number 1.
L184-185. The authors do not collect whether the differences between boys and girls are statistically significant in the whole test and the subtests.
-Table 1 is very confusing, could you please consider simplifying or at least put firstly the results of the dictionary Test for the studies children? The data of the results of the dictionary test for the studied children should be presented before what the results of subgroups.
L204: The value should be checked because the 1.9 standard deviation corresponds to “creating subordinate words” and not to “creating superordinate words”
L207: “the correlation between the level of the dictionary of the studied population and age in the whole test and individual sub-test varied (Table 1.)” However, the Table 1 only collects the mean scores and standard deviation. This information is missing. It is important that the authors discuss this information because there should not be a correlation between age and speech deficits.
What is the purpose of dividing the participants by age groups, of later this score is not used to relate it to the scores on the CTSIB Test?
L232. The sentence “children with a low level of vocabulary in the general evaluation or the closed eyes hard surface Dictionary Test obtained significantly worse results than the group with an average dictionary level” does not match the results in the Table 3. The low group mean is 2.51 and the average group mean is 2.34.
L242. The information in Table 3 could be presented in a simplified version if the authors do not repeat the number of participants column. In addition, it is not clear if the p-value refers to total participants or by levels. If it was the latter case, the authors must put the p-value for each level.
L240. Is compound result or complex result?
DISCUSSION
L258. I do not understand the relationship between deficits in imitation of movements and speech deficits in the vocabulary. The authors could explain the importance of this relation in their study.
-L273-L290. “It is in the 5th year…” This statement is not supported by any data or reference.
-L297-298. The children in Góral et al. (2014) study is developmentally or language impaired children? This is important in order to discuss the results their study. Additionally, the authors present many results of the study by Góral et al. (2014) that are not discussed with the results of their research. Moreover, the age variable could influence the results because Góral's study is with 3-year-old children and the age of the children is 5 years in this study.
L341. What kind of motor disorders present the children in Romaniec´s study?
-Overall, I would suggest the authors to reconsider writing this section from scratch. The discussion is mostly speculative and a little disorganized; an interpretation of the results has not been presented; and limitations of the study have not been highlighted.
Conclusions
L395. The authors should clarify this sentence “This research may contribute to more reliable diagnosis of children with speech deficit in terms of dictionary and sensory integration and balance” because the aim of their study is to assess the relationship between these variables and they only observed statistically significant results in closed eyes condition.
The authors could explain more extensively the implications of their study.
Author Response
- I am submitting a revised article, "Speech Development Deficits in Terms of Dictionary and Processes of Sensory Integration and Balance in Five-Year-Old Children” to be considered for publication in Children. I confirm that this work is original and has not been previously published, nor is it currently under consideration for publication elsewhere. The problem addressed in this work is relevant to the scope of your journal. All procedures performed in studies involving human participants were in accordance with the ethical standards of the institutional and/or national research committee and with the 1964 Helsinki declaration and its later amendments or comparable ethical standards. We would like to thank the reviewers for the time and effort put into the review of our manuscript. It was quite a challenge for us to answer some of the greatly insightful and very detailed questions. However, we have made every effort to meet this challenge. We hope that after undergoing revision and following extensive changes, the article will prove interesting and will be accepted for publication. Below, please find a detailed, point by point, description of the changes applied in the text as well as responses to comments.
- Response to comments by Reviewer 2
- Thanks very much for asking me to review the manuscript “Speech development deficits in terms of dictionary and processes of sensory integration and balance in five-year-old children”, submitted for publication in Children. The authors have conducted a study to evaluate the relationship between speech deficits in terms of dictionary as well as sensory integration and balance in children. The study is overall interesting, however, after careful review, I am advising the editors to re-consider the manuscript for publication after major revisions and resubmission by the authors. Introduction, methods and discussion should be extensively refined and improved.
- Point 1: INTRODUCTION
- L26. The authors make a differentiation between language and speech. If they consider that they have to make such a differentiation, they must take into account that language is the ability of humans to express and communicate desires, feelings and thoughts through a system of particular signal like sounds, voice, written symbols and gestures. While the speech I the act of the expressing or the faculty of describing desires, feelings and thoughts by words or vocal communication.
- Response 1:
- In the Introduction, I included the sentence: Language is people's ability to express and communicate their desires, feelings, and thoughts through a system of specific signals, such as sounds, voice, written symbols, and gestures. Whereas speech is the act of expressing or being able to describe desires, feelings and thoughts in words or by voice communication.
- L41. The sentence “Motor development affects the processes of thinking and processing stimuli reaching the child from the environment” would benefit from a supporting reference.
- L41. After the sentence: „Motor development affects the processes of thinking and processing stimuli reaching the child from the environment” i put a reference: Westendorp M, Hartman E, Houwen S, Smith J, Visscher C. The relationship between gross motor skills and academic achievement in children with learning disabilities. Res Dev Disabil. 2011 Nov-Dec;32(6):2773-9. doi: 10.1016/j.ridd.2011.05.032.
- L46. The authors should concrete the sentence “At the age of 6, a child affected by developmental deficits will have problems…” There are no defined the “developmental deficits”.
- L46. I corrected that sentence: An important determinant is the condition of the small motor skills, oral praxis, modulation of muscle tone throughout the body and planning its movements. Disorders in this regard can be observed before the first year of life. In most children, the above symptoms only mean transient developmental difficulties. However, some of them, aged 6-7 years old, may experience problems with modulating stimuli coming from the equilibrium system. This can be seen by watching a child play and the inharmonious development of particular manipulative and motor skills [13].
- In the discussion appear a series of research that address the aim of this investigation but that do not appear in the introduction. These investigations should be mentioned in the introduction. Also, all investigations that appear in the introduction are not discussed in the discussion section. It does not appear that there is a relationship between the state of the art and the discussion of the results of this research.
- In the Introduction, there are now works cited in the Discussion.
- The authors must define what they understand by children with speech deficits: SLI, phonological disorders, developmental co-ordination disorder (dispraxya), etc.
- We resigned from the term - speech development deficits in the dictionary aspect, both in the title and in the rest of the article. Our research focused on the level of development of a five-year-old child's dictionary by gender. The assessment of the level of dictionary development included the assessment of the child's word formation level, the assessment of the concept definition and the assessment of parental word formation. Dictionary development was assessed as low, medium or high.
- The aim is to assess the relationship between speech deficits regarding dictionary and sensory integration balance in 5-year-old children without differences by sex. However, the results are explained in terms of boys and girls. The aim should be to rewrite.
- We changed the aim of the research to: The aim of the research was to assess the relationship between the level of dictionary development and selected elements of sensory integration and balance of five-year-old girls and boys.
- On the other hand, the authors do not explain the hypothesis or hypotheses. Z drugiej strony autorzy nie wyjaśniają hipotezy ani hipotez.
- We clarified the hypotheses: Hypothesis: We assumed that there is a significant relationship between the level of vocabulary development and selected elements of sensory integration and balance. The greater the disturbance in the processes of sensory integration and balance, the lower the child's dictionary development.
- Point 2.
- METHODS
- There are several issues in the method that need to be carefully addressed.-The patient selection: The description of the participants is scarce. We do not know if all the participants have development deficits of speech and disturbances of sensory integration and balance processes at the same time, what kind of speech development deficits have the participants. The manuscript does not explain if the participants attend a special education school, who has diagnosed them, what is the degree of speech development deficits and disturbances of sensory integration and balance processes. They authors could provide more information on participants and define them specifically because we do not know if they have speech delay, language delay or SLI.
- I corrected some of the work Material and methods. The study group consisted of 290 children aged 5 (172 boys and 118 girls). The examined children attended municipal kindergartens. The level of dictionary development of the respondents was assessed as low, medium or high. Most of the children experienced disturbances in selected elements of sensory integration and balance. This is presented in Tables 1 and 2. Criteria for inclusion in the study: five-year-olds, girls and boys, presence of selected disorders of sensory integration processes (SI), consent to the study. Exclusion criteria: age under and over five years of age, physiological hearing impairment, intellectual disability, diseases and congenital defects of the central nervous system. Before the tests were performed, the following documents were prepared: information for the test taker containing detailed data on the purpose and procedures of the tests; form of consent of the parent / legal guardian for the child's participation in the study; declaration of the child's parent / legal guardian on the processing of data related to the participation in the study; a statement in which the researcher undertakes to obtain informed consent for the study from all parents / legal guardians of children participating in the study. The research was conducted in the first half of 2016 in the Posturology Laboratory of the Collegium Medicum of the Jan Kochanowski University. All research procedures were carried out in accordance with the Helsinki Declaration of 1964 and with the consent of the University Bioethics Committee for Research at Jan Kochanowski University in Kielce (Resolution No. 23/2015).
- Outcome measures: L113: It is important that authors explain what the range of scores “low, average and high” to interpret the results. In addition, the authors must report on how the CTSIB is scored. In the procedure, it would be interesting for the authors to explain how the evaluation was, individually, at the participants' home, in the hospital office, how long did the assess last with each participant, firstly, which they assessed first, the language component or the motor component, etc. - Mierniki rezultatu: L113:
- Corrected. The research was carried out in the first half of 2016 in the Laboratory of Posturology, Collegium Medicum, Jan Kochanowski University.
- Point 3. RESULTS
- Table 1 is very confusing, could you please consider simplifying or at least put firstly the results of the dictionary Test for the studies children? The data of the results of the dictionary test for the studied children should be presented before what the results of subgroups. L204: The value should be checked because the 1.9 standard deviation corresponds to “creating subordinate words” and not to “creating superordinate words” L207: “the correlation between the level of the dictionary of the studied population and age in the whole test and individual sub-test varied (Table 1.)” However, the Table 1 only collects the mean scores and standard deviation. This information is missing. It is important that the authors discuss this information because there should not be a correlation between age and speech deficits.What is the purpose of dividing the participants by age groups, of later this score is not used to relate it to the scores on the CTSIB Test? L209. Please, consider rearranging the results. First, they could explain the total results and then the results by gender. Second, the total results in the subtest and then the results by gender in them. In this way they avoid the repeated results that appear in the different lines (i.e., L167 and L177, 3.71). The authors should change the abbreviatures of mean (X) and standard deviation (s). Mean= M and Standard Deviation= SD. The p-values must be written in lowercase in the Table 2.
- I corrected table 1 ,2 i 3 and its description.
- L232. The sentence “children with a low level of vocabulary in the general evaluation or the closed eyes hard surface Dictionary Test obtained significantly worse results than the group with an average dictionary level” does not match the results in the Table 3. The low group mean is 2.51 and the average group mean is 2.34.
- L242. The information in Table 3 could be presented in a simplified version if the authors do not repeat the number of participants column. In addition, it is not clear if the p-value refers to total participants or by levels.
- If it was the latter case, the authors must put the p-value for each level.
- L240. Is compound result or complex result?
- Response 2:
- I corrected table 1 ,2 i 3 and its description.
- Point 3.
- DISCUSSION
- L258. I do not understand the relationship between deficits in imitation of movements and speech deficits in the vocabulary. The authors could explain the importance of this relation in their study. L258. L273-L290. “It is in the 5th year…” This statement is not supported by any data or reference. L273-L290. „Jest w piÄ…tym roku…” L297-298. The children in Góral et al. (2014) study is developmentally or language impaired children? This is important in order to discuss the results their study. Additionally, the authors present many results of the study by Góral et al. (2014) that are not discussed with the results of their research. Moreover, the age variable could influence the results because Góral's study is with 3-year-old children and the age of the children is 5 years in this study.-L297-298.L341. What kind of motor disorders present the children in Romaniec´s study?L341.Overall, I would suggest the authors to reconsider writing this section from scratch. The discussion is mostly speculative and a little disorganized; an interpretation of the results has not been presented; and limitations of the study have not been highlighted.
- Response 3:
- The Discussion has been corrected
- Pont 4.
- Conclusions
- L395. The authors should clarify this sentence “This research may contribute to more reliable diagnosis of children with speech deficit in terms of dictionary and sensory integration and balance” because the aim of their study is to assess the relationship between these variables and they only observed statistically significant results in closed eyes condition. The authors could explain more extensively the implications of their study.
- Response 4.
- The conclusions were corrected
- Most of the children had a medium to high level of dictionary development. Less developed vocabulary skills to Creating subordinate words and Defining concepts. There were no significant differences in the level of the dictionary between girls and boys. We observed disorders of selected elements of sensory integration and balance in most children and more often in boys. There were statistically significant relationships between the level of the dictionary and selected disorders of sensory integration and balance, but they were not unambiguous. Children with the lowest level of the dictionary in Overall assessment obtained significantly the worst results in the CTSIB Open eyes, hard surface test. However, in the Eyes closed, hard surface test, the lowest score was obtained by children with a high Overall assessment. In turn, in the Open eyes, soft surface test, the lowest score was obtained by children with an average Overall assessment. In the CTSIB - Complex result test, the lowest score was obtained by children with Low Defining concepts.
- Once more, we are exceptionally grateful for your in-depth view of our article. Your insight and comments will definitely allow for an increase in the substantive value of the manuscript. We hope that our detailed responses and the extensive changes to the text are sufficient for the publication of our text. Thank you for your devoted time and effort.
Yours sincerely,
Assoc. Prof. UJK Jacek Wilczyński, Ph.D.

Round 2
Reviewer 2 Report
INTRODUCTION
L109: There is a typo “..the occurrenc e of”
The aim is to assess the relationship between speech deficits regarding dictionary and sensory integration balance in 5-year-old children without differences by sex. However, the results and the discussion are explained in terms of boys and girls. The aim should be to rewrite.
In the first review, I account for the authors do not explain in the aim of the study that they are going to study the difference between boys and girls. This is still not included in the objective of the manuscript. If the authors do not include this information, they should modify the presentation of the results.
METHODS
-The patient selection: The authors account for in the manuscript “The majority of children had a medium to high level of dictionary development”. However, in the results section the participants have “a low level of dictionary development”. It seems contradictory.
L144: There is a typo “and thevelopment of the”.
L210. The format of the Table 1 should be reviewed.
RESULTS
L203. The authors should check English …”had a low dictionary level, 108 had average (37%) and high 64 (22%).
-L221, L223, L226, L228, L283. In some p-values results is omitted the point “…here (p = 018)”.
L240. The information in Table 3 could be presented in a simplified version if the authors do not repeat the number of participants column.
DISCUSSION
L320. In Jodzis´s research “There were also differences in the vocabulary of 5-year-old boys and girls. The general dictionary development of girls turned out to be higher than boys”; What would be the explanation for why in this research the result is different: girls have worse scores than boys in the total test? The authors should explain it.
- The limitations of the study have not been highlighted.
CONCLUSIONS
L357. “The majority of children had a medium to high level of dictionary development” This conclusion does not fit the results presented in Table 1 because 50% of the population (Med) have a score of 3 which would correspond to a low level, according to the range explained by the authors.
The authors could explain more extensively the implications of their study.
Author Response

(The authors gave the same response as above.)
